# Longitudinal Dynamics of SARS-CoV-2 IgG Antibody Responses after the Two-Dose Regimen of BNT162b2 Vaccination and the Effect of a Third Dose on Healthcare Workers in Japan

**DOI:** 10.3390/vaccines10060830

**Published:** 2022-05-24

**Authors:** Atsuhiko Sakamoto, Michinobu Yoshimura, Ryota Itoh, Ryo Ozuru, Kazunari Ishii, Yusuke Sechi, Shigeki Nabeshima, Kenji Hiromatsu

**Affiliations:** 1Department of Microbiology & Immunology, Faculty of Medicine, Fukuoka University, Fukuoka 814-0180, Japan; a.sakamoto.sw@fukuoka-u.ac.jp (A.S.); myoshimura@fukuoka-u.ac.jp (M.Y.); ryito@fukuoka-u.ac.jp (R.I.); ozuru@fukuoka-u.ac.jp (R.O.); kishii@fukuoka-u.ac.jp (K.I.); 2General Medicine, Fukuoka University Hospital, Fukuoka 814-0180, Japan; yusuke1992@adm.fukuoka-u.ac.jp (Y.S.); snabeshi@fukuoka-u.ac.jp (S.N.)

**Keywords:** mRNA vaccine, antibody response, SARS-CoV-2, BNT162b2

## Abstract

Analysis of longitudinal dynamics of humoral immune responses to the BNT162b2 COVID-19 vaccine might provide useful information to predict the effectiveness of BNT162b2 in preventing SARS-CoV-2 infection. Herein, we measure anti-RBD IgG at 1, 3 and 6 months (M) after the second dose of BNT162b2, and at 1 M after a third dose of BNT162b2 vaccination in 431 COVID-19-naïve healthcare workers (HCWs) in Japan. All HCWs mounted high-anti-RBD IgG responses after the two-dose regimen of BNT162b2 vaccinations. Older persons and males presented lower anti-RBD IgG responses than younger adults and females, respectively. The decay in anti-RBD IgG started from 1 M after the second dose of BNT162b2 and anti-RBD IgG titers dropped to nearly one-tenth at 6 M after the second vaccination. Subsequently, the participants received a third dose of BNT162b2 at 8 M after the second dose of BNT162b2 vaccine. Anti-RBD antibody titers 1 M after the third dose of BNT162b2 increased seventeen times that of 6 M after the second dose, and was twice higher than the peak antibody titers at 1 M after the second dose of vaccination. The negative effect of age for the male gender on anti-RBD IgG antibody titers was not observed at 1 M after the third dose of BNT162b2 vaccine. There were no notable adverse events reported, which required hospitalization in these participants. These results suggest that the third dose of BNT162b2 safely improves humoral immunity against SARS-CoV-2 with no major adverse events.

## 1. Introduction

The BNT162b2 (Pfizer-BioNTech) and mRNA-1273 vaccines (Moderna), which encode a modified SARS-CoV-2 full-length spike protein, have been widely used as vaccines against SARS-CoV-2 and have been reported to be highly effective in preventing SARS-CoV-2 infection, hospitalization, and even death [1,2,3,4,5]. However, waning humoral immunity to BNT162b2 vaccination over time and the prevalence of variant SARS-CoV-2 that are less sensitive to the antibodies induced by vaccination contribute to the increasing appearance of breakthrough infections [6,7]. Hatzakis et al. reported the profound decline in the anti-receptor binding domain (RBD) antibody 4–8 months after the first dose of BNT162b2 is a harbinger of a reduction in vaccine effectiveness to prevent SARS-CoV-2 infection [8]. Thus, it is important to know the longitudinal dynamics of humoral responses to SARS-CoV-2 vaccination to predict the effectiveness of BNT162b2 vaccine in preventing SARS-CoV-2 infection.

Levin et al. reported that 6 months after the two-dose regimen of COVID-19 mRNA vaccinations, humoral IgG responses were substantially decreased, especially among men, among persons 65 years of age or older and among those with immunosuppression [9]. Israel et al. also reported that antibody titers decreased by up to 38% each subsequent month, and 6 months after vaccination, 16.1% subjects had antibody levels below the seropositivity threshold in vaccinated subjects [10]. On the other hand, several reports demonstrated the antibody persistence at six months, albeit with a certain decline. Yet, the titer was still one order of magnitude higher than that before vaccination [11,12,13]. Thus, long-term kinetics data of the humoral responses after the two-dose regimen of COVID-19 mRNA vaccines are still controversial.

Here, we report on humoral immunity from 1 month to 6–7 months after the second dose of BNT162b2 in 431 Japanese HCWs without previous or de novo infection with SARS-CoV-2. We additionally analyzed anti-RBD IgG antibody titers 1 M after the third dose of BNT162b2 vaccination. We demonstrate that the third dose (booster shot) of BNT162b2 vaccination strikingly augments anti-RBD IgG responses of those mild responders, namely, older persons and males. Collectively, this study confirms key groups that may benefit from additional vaccine doses when available.

## 2. Materials and Methods

### 2.1. Study Population and Survey Period

Participants were recruited from the HCWs of Fukuoka University Hospital who received the two doses of BNT162b2 vaccination. The requirements for participation were as follows: (1) aged 20 years or older; (2) no malignancy, liver disease, renal disease, diabetes mellitus, immunodeficiency, or other diseases that may cause immune dysfunction; and (3) no use of immunosuppressive drugs. Those who were positive for anti-SARS-CoV-2 S-protein RBD antibodies (three participants) or anti- nucleocapsid (N)-protein antibodies (two participants) before vaccination were excluded. Informed consent was obtained from all the participants. The participants’ background information, such as age, gender, height, weight, history of COVID-19, and chronic diseases, were collected using a questionnaire. Among the participants, 337 (78.2%) were female (median: 41.0 years of age; IQR: 30.0–49.0) and 94 (21.8%) were male (median: 37.5 years of age; IQR: 32.0–48.8) (Table 1).

A large portion of our cohort individuals received a third dose of BNT162b2 vaccination from 11 to 21 January 2022 at around 8 M after the second dose of BNT162b2 vaccination, in accordance with the Japanese Ministry of Health, Labor and Welfare recommendation to for healthcare workers. Blood samples were first collected from 8 to 18 March 2021 (before the first BNT162b2 vaccination) as a pre-vaccination survey, from 31 May to 14 June (corresponding to approximately 28–56 days after the second dose (2D-1M)), from August 3 to 13 (corresponding to approximately 90–120 days after the second dose (2D-3M)), from 8 to 17 November 2021 (corresponding to approximately 180–210 days after the second dose (2D-6M)), and from 21 February to 4 March 2022 (corresponding to approximately 28–56 days after a third dose of BNT162b2 vaccine (3D-1M)) (Figure 1).

### 2.2. ELISA for Anti-SARS-CoV-2 RBD IgG or Anti-N-Protein IgG Detection

The spike (S)-protein receptor-binding domain (RBD) vector (NR-52309) and full-length SARS-CoV-2 N-protein expression vector (NR-53507) were kindly provided by BEI Resources (Manassas, VA, USA). Recombinant protein of the S-protein RBD region and N-protein were expressed and purified based on a previously reported method [14]. The ELISA protocol was adapted from previously established protocols [15]. Overnight, 96-well plates for ELISA (Thermo Scientific TM 442404, Waltham, MA, USA) were coated at 4 °C with 50 μL per well of 2 μg/mL recombinant RBD protein or 1 μg/mL recombinant N protein. The plates were then blocked for 2 h at room temperature with 100 μL of 3% *w*/*v* skim milk (BD Difco^TM^ 232100, Waltham, MA, USA). Diluted serum samples were added to the plate (50 μL/well) and incubated at 37 °C for 2 h. Then, after washing again, 50 μL of HRP-labeled goat anti-human IgG antibody (Jackson ImmunoResearch 109-035-008 (West Grove, PA, USA); diluted to 1:5000) was added as a secondary antibody and incubated at 37 °C for 2 h. On washing, 50 μL of TMB substrate (BioLegend 421101, San Diego, CA, USA) was added and allowed to react for 5 min, then the reaction was stopped with 50 μL of 2 N sulfuric acid solution, and the absorbance, at 450 nm, was measured using a plate reader (Bio-Rad iMARKTM microplate reader, Hercules, CA, USA). The absorbance of the blank well was subtracted and used as absorbance450 (ABS_450_) for the analysis. For the anti-SARS-CoV-2 N-IgG antibodies, we measured the ABS_450_ of serum, which was diluted 400 fold. For anti-SARS-CoV-2 S-RBD IgG, serum samples were assayed at a 1:200 starting dilution and 6 additional 4-fold serial dilutions. The absorbance sum across 6 additional 4-fold serial dilutions was used as SUM. This SUM calculation was modified from the method described by Weisberg et al. [16]. Furthermore, monoclonal antibody equivalents were determined to quantitatively evaluate the attenuation of antibody titer over time. Human anti-SARS-CoV-2-S1 monoclonal IgG (Abcam, ab273073, Waltham, MA, USA) at seven different concentrations of 0.5 ng/mL to 8.0 ng/mL were simultaneously measured as reference standards in each plate, and the ABS450 data of samples between 0.2 and 0.7 were used to calculate the regression line between antibody concentration and ABS_450_ values. Then, for each serum sample with ABS_450_ between 0.2 and 0.7 in the stepwise dilutions, (equivalent concentration (ng/mL) of monoclonal antibody deduced from the regression line) * dilution factor/200 was calculated to be the arbitrary unit (A.U.) of that sample. If the ABS_450_ values of the samples in multiple dilution series steps fell within the range of 0.2 to 0.7, the average of the A.U. of those samples was used as the A.U. of the serum in consideration.

### 2.3. Statistical Analyses

A paired samples t-test was used to compare antibody titers before and after vaccination. For comparisons between sexes and between BMI groups, an unpaired t-test was performed. Analysis of variance was performed for comparisons by age group and smoking status, while Tukey’s method was used to test for significant differences between groups. At the same time, linear regression analysis was performed to determine the 95% confidence interval of the slope. Wilcoxon matched-pairs signed rank test was used to compare 2D-1M antibody titers with 3D-1M antibody titers by strata. Multiple regression analysis was performed on variables that were significant in the univariate analysis. These statistical analyses were performed using Microsoft Excel, GraphPad Prism and R software version 4.1.2 (R foundation). A *p* < 0.05 was considered statistically significant.

## 3. Results

### 3.1. Demographic Data

Out of a total of 1956 employees from Fukuoka University Hospital, 1666 received the first mass vaccination from March–April 2021. The current study enrolled 457 participants and we excluded 26 individuals by the following reasons, such as lack of data, anti-spike or anti-N-protein seropositivity or incomplete vaccine inoculations, leaving a total of 431 subjects for analysis (Figure 1). At the start of the third dose, the 431 participants in the analysis were invited to the longitudinal prospective antibody kinetic study after the third dose of the BNT162b2 vaccine, and 370 were enrolled. Of these, ten subjects were excluded from the 3D-1M analysis because of seropositivity for N antibodies and three hundred and sixty HCWs were included in the 3D-1M analysis (Figure 1).

Antibody titers were analyzed in 412, 405 and 394 blood samples collected at 28–56 days (2D-1M), 90–120 days (2D-3M) and 180–210 days (2D-6M) after the second dose of the BNT162b2 vaccination, respectively. Antibody titers at 28–56 days (3D-1M) after the third dose of BNT162b2 vaccination from 360 subjects were also analyzed.

### 3.2. Measurement of Anti-SARS-CoV-2 N-Protein IgG and Anti-SARS-CoV-2 S-Protein RBD IgG by In-House ELISA

We confirmed the purity of recombinant SARS-CoV-2 N protein by SDS-PAGE (Figure 2A). For anti-N IgG ELISA, we decided to measure the ABS_450_ value of serum diluted 400-fold, based on the results of stepwise dilutions of negative and positive controls (Figure 2B). The ABS_450_ value of the anti-N IgG antibody in 100 negative controls had a mean of 0.216 with a standard deviation of 0.135 (Figure 2C). Using the mean plus 3 times the standard deviation as the cutoff, the cutoff value was 0.620. On the other hand, the ABS_450_ of 29 positive controls collected from COVID-19 patients more than 14 days after the onset was above the cutoff value of 0.620 in all cases. The sensitivity and specificity indices calculated on this basis were 98% and 100%, respectively.

We also confirmed the purity of recombinant SARS-CoV-2 RBD protein by SDS-PAGE (Figure 2D). For the measurement of anti-SARS-CoV-2 RBD-IgG antibodies, we decided to measure the ABS_450_ value of serum diluted 200-fold, based on the results of stepwise dilutions of negative and positive controls (Figure 2E). The ABS_450_ value of the anti-RBD antibody in 199 negative controls had a mean of 0.160 with a standard deviation of 0.063, resulting in a cutoff value of 0.348 when the mean plus 3 times the standard deviation was used as the cutoff (Figure 2F). On the other hand, the ABS_450_ of 14 positive controls collected from COVID-19 patients more than 14 days after the onset was above the cutoff value of 0.348 in all cases. The sensitivity and specificity indices calculated based on the above were 99% and 100%, respectively.

### 3.3. Kinetics of Anti-RBD IgG Titers after the Second Dose and the Third Dose of BNT162b2 Vaccination in HCWs in Japan

Measurements of IgG specific for N protein, which is not encoded in the vaccine, revealed that two participants were positive prior to vaccination (Figure 3A). There was one participant seroconverted for anti-N IgG 3 M after the second dose of BNT162b2 vaccine. As previously mentioned, these three individuals were excluded from the following analyses. Ten participants were found to be positive for anti-N IgG at 1 M after the third dose of BNT162b2 (Figure 3A, 3D-1M). Among these ten seroconverted participants, nine were diagnosed as COVID-19 either by SARS-CoV-2 real-time RT-PCR or the SARS-CoV-2 antigen detection kit, and none of them required hospitalization. All these anti-N IgG seroconverted individuals were excluded from the following analyses of anti-Spike RBD IgG after the third dose of BNT-162b2 vaccination (3D-1M).

At 1 M after the second BNT162b2 vaccination, all subjects acquired robustly high titers of anti-RBD IgG antibody, which peaked around day 28 after the second vaccination and steadily decreased over the following 6-month period (Figure 3B–E). Our cohort individuals received the third dose of BNT162b2 vaccination at around 8 M after the second dose of BNT162b2 vaccination, in accordance with the Japanese Ministry of Health, Labor and Welfare recommendation to health care workers. We found that anti-RBD IgG antibody responses drastically improve after a third-dose vaccination (Figure 3F). We noticed that the post-vaccination serum samples at 1 M after the second dose of vaccine (2D-1M), 3 M after the second dose of vaccine (2D-3M), and 1 M after the third vaccination (3D-1M) contain very high anti-RBD IgG titers and the 200-fold or 400-fold dilutions were considered insufficient and inappropriate for the quantitative analyses because ABS_450_ value at these ranges of dilution already reached the plateau and the resolution was poor for the analyses (Figure 3C,D,F). Therefore, the sum of ABS_450_ values of six serial 4-fold dilutions (SUM) was used for the longitudinal kinetics study of anti-RBD IgG antibody.

### 3.4. Longitudinal Kinetics of Anti-SARS-CoV-2 RBD IgG Antibodies after the Second and Third Doses of BNT162b2 Vaccine

Longitudinal kinetic of humoral antibody immune responses to BNT162b2 over 6 months after the receipt of the second vaccination and the effect of a third-dose vaccine on anti-RBD IgG responses were examined using SUM. Regression analysis of RBD antibody titers (SUM) of all 1252 samples from 431 subjects after two doses of vaccination showed a significant decrease in antibody titer over time (Figure 4A). We found that the anti-RBD IgG antibody titer peaked at around 1 M after the second vaccination, followed by a substantial reduction in the IgG levels over time. We observed striking increases in anti-RBD IgG antibody levels at 1 M after a third dose (booster shot) of BNT162b2 (Figure 4B). To further estimate the dynamics of anti-RBD IgG antibody titers, we also measured A.U., which is proportional to the equivalent dose of anti-RBD monoclonal antibodies (IgG), of the serum 28–56 days (2D-1M), 90–120 days (2D-3M) and 180–210 days (2D-6M) after the second dose (Figure 4C). The median antibody titers (A.U.) at 1 M, 3 M and 6 M after the second dose was 129.9 (Interquartile Range (IQR): 90.1–205.8), 38.7 (IQR: 26.0–66.5) and 17.5 (IQR: 9.9–27.0), respectively. The anti-RBD IgG level of 2D-6M was 13.5% of 2D-1M. Strikingly, anti-RBD IgG levels greatly increased at 1 M after the third dose of BNT162b2 and the median antibody titer at 1 M after the third dose was 296.9 (IQR: 192.0–476.7) (Figure 4D). Thus, the antibody levels of 3D-1M were a 17-fold increase compared to those of 2D-6M, and a 2.2-fold increase in the peak of the two-dose BNT162b2 vaccinations (2D-1M). These results strongly suggest that a third dose of BNT162b2 greatly improves humoral immunity against SARS-CoV-2.

Interestingly, by using the antibody titer at 2D-1M as the peak of the second dose vaccination, the attenuation rate was 70.2% at 3 M and 86.5% at 6 M. We fitted a linear regression model with R software version 4.1.2 (R foundation) to quantify the association between days since the second dose of vaccination and the logarithm of antibody levels. The F-test showed that the number of days that elapsed after the administration had a significant negative effect on antibody titers (*p* < 0.001). The half-life of antibody titers at 3 M from the peak humoral response (1 M) was calculated from the regression coefficients to be 37.42 days (95% CI: 34.80–40.46) (Figure 4C). The decrease from the start of 3 M onward was much slower and the half-life of antibody titers at 6 M from 3 M was calculated from the regression coefficients to be 72.89 days (95% CI: 66.23–81.02), indicating the decrease in antibody titers was initially brisk, in the period of up to 3 M after the second-dose vaccination, but slowed and persisted thereafter. It is important to note that this is an ongoing longitudinal prospective cohort study and it will be very important to examine the trajectories of anti-RBD IgG (declining slope) at 1 M, 3 M and 6 M after the third dose of the BNT162b2 mRNA vaccine, so that we can predict the effectiveness of the BNT162b2 vaccine in preventing SARS-CoV-2 infection.

### 3.5. Third Dose of BNT162b2 Vaccination Augments Anti-RBD IgG Responses of Mild Responder at 1 M after the Second Dose of BNT162b2 Vaccination

Cohort participants were divided into three groups based on anti-RBD IgG titers (SUM) at 2D-1M, namely, mild responder: <mean −1 SD, moderate responder: mean ± 1 SD and high responder: mean + 1 SD≤ (Figure 5A). Anti-RBD IgG titers at 1 M after the third dose of BNT162b2 vaccination (3D-1M) were compared with the titer of 2D-1M in these three groups (Figure 5B and Appendix A). Noteworthily, a third dose (booster shot) of BNT162b2 vaccination strikingly augmented the anti-RBD IgG responses of mild responders (Figure 5B). On the other hand, the same booster shot restored anti-RBD IgG responses of high responders, but did not enhance the anti-RBD IgG levels of 2D-1M further (Figure 5D), indicating that anti-RBD IgG responses of high responders might have already reached the plateau after the two-dose regimen of BNT162b2 vaccination. The demographic characteristics of these three responder groups (mild, moderate and high) based on reactivity at 2D-1M are shown in Table 2. Based on these demographic characteristics, we next moved on to further analyses to determine what influences anti-RBD IgG reactivity after the 2-dose regimen and a third-dose BNT162b2 vaccination in our cohort HCWs without previous or de novo infection with SARS-CoV-2 in Japan.

### 3.6. Negative Effect of Age on Anti-RBD IgG Responses after the Second Dose of BNT162b2 Vaccination Was Lost after the Third Dose of BNT162b2 Vaccination

Among the participants, 337 (78.2%) were female (median: 41.0 years of age; IQR: 30.0–49.0) and 94 (21.8%) were male (median: 37.5 years of age, IQR: 32.0–48.8). The stratified analysis by age groups (in 10-year bins) of the HCW cohort revealed an increasingly negative effect of age at 1 M after the second vaccination (2D-1M) (Figure 6A). The participants whose ages were between 23 and 34 years had higher antibody titers than those age groups from 45 to 54 years and 55 to 64 years. In addition, regression analysis indicated that the older the age, the lower the antibody titer (Figure 6C, 2D-1M shown in red). Furthermore, the inverse correlation between age and antibody titer observed at 1 M after the second does was maintained at 3 M and 6 M after the second dose as well (Appendix A). There was no significant difference in the rate of antibody decay with age (Appendix A). On the contrary, the negative effect of age on anti-RBD IgG responses to BNT162b2 observed at 1 M, 3 M and 6 M after the second dose of BNT162b2 vaccine was lost 1 M after the third dose (3D-1M) of BNT162b2 vaccination, and there was no difference in antibody titer among age groups at 3D-1M (Figure 6B). Furthermore, regression analysis showed that the slope of 2D-1M and that of 3D-1M statistically differs indicating booster shots of third-dose BNT162b2 mRNA vaccine specifically enhances anti-RBD IgG responses of aged individuals and the negative impact of age on antibody responses was no longer observed (Figure 6C, 3D-1M shown in blue).

### 3.7. Effect of Gender and BMI on Anti-RBD IgG Responses at 3 M and 6 M after the Second Dose and 1 M after the Third Dose of the BNT162b2 Vaccination

Anti-RBD IgG antibody titers in serum acquired at 1 M after the second dose of BNT162b2 vaccine were significantly higher in females (4.305 SUM (IQR: 3.930–4.627)) than in males (4.180 SUM (IQR: 3.704–4.552)) (Figure 7A, 2D-1M *p* = 0.0233). However, the rate of decrease in the antibody titer between 2D-3M to 2D-6M period was faster in females than in males (Appendix A), and the gender differences observed 1 M after vaccination were not detected at 3 M and 6 M after the second dose (Figure 7A, 2D-3M, 2D-6M). Surprisingly, the antibody titers at 1 M after the third dose of the BNT162b2 vaccine (3D-1M) were conversely higher in males (4.833 SUM (IQR: 4.390–5.280) than in females (4.581 (IQR: 4.111–5.070)) (Figure 7A, 3D-1M, *p* = 0.0226).

In the analysis based on BMI, antibody titer at 1 M after the second dose of vaccination was significantly higher in obese subjects with BMI ≥ 30 (4.616 SUM (IQR: 4.349–5.417)) than in those without obesity BMI < 30 (4.272 (IQR: 3.889–4.617)) (Figure 7B, 2D-1M). Although no difference in antibody titer based on BMI was observed at 3 M or 6 M after the second vaccination in our cohort, antibody titers at 1 M after the third dose of vaccine was again higher in obese subjects with BMI ≥ 30 (5.470 SUM (IQR: 4.868–6.148) than in those without obesity (4.600 SUM (IQR: 4.165–5.096)) (Figure 7B).

Regarding smoking status, there was a tendency for current smokers and ex-smokers to have lower antibody titers than never-smokers, but the difference was not statistically significant (Figure 7C).

### 3.8. Multivariate Analysis on the Effects of Sex, Age and BMI on Anti-RBD IgG Responses at 1 M after the Second Dose and 1 M after the Third Dose of BNT162b2 Vaccinations

Based on the results mentioned above, and to compare 2D-1M and 3D-1M, multiple regression analysis was performed with antibody titer (SUM) as a response variable and age, sex and BMI as explanatory variables (Table 3). Since gender is a categorical variable, a dummy variable was fitted with females as 0 and males as 1. Focusing on the standard partial regression coefficients (in parenthesis), age (−0.230, *p* < 0.001) contributed the most in 2D-1M, followed by gender (−0.157, *p* = 0.004) and BMI (0.106, *p* = 0.054). The regression equation in 2D-1M is as follows (F = 9.166, *p* < 0.001).
$$\left[\mathrm{IgG}\;\mathrm{titer}\;\mathrm{in}\;2D\text{-}1M\right]=4.380-0.013\times \left[\mathrm{Age}\right]-0.230\times \left[\mathrm{Sex}\right]+0.022\times \left[\mathrm{BMI}\right]$$

In contrast, at 3D-1M, BMI made a stronger (0.134, *p* = 0.017), gender made a weaker (0.096, *p* = 0.083), and age made little or no (−0.044, *p* = 0.412) contributions, respectively. The selection of explanatory variables based on the Akaike Information Criterion (AIC) yielded a regression model that can be explained only by BMI and gender, and the regression equation is as follows (F = 5.619, *p* = 0.004).
$$\left[\mathrm{IgG}\;\mathrm{titer}\;\mathrm{in}\;3D\text{-}1M\right]=3.958+0.158\times \left[\mathrm{Sex}\right]+0.029\times \left[\mathrm{BMI}\right]$$

Since age, the most critical variable in 2D-1M, is no longer significant in 3D-1M, these results indicate that the age factor disappears after the three-dose vaccination because the IgG titer of mild responders (namely, those who are the older, in Figure 6) significantly increases.

## 4. Discussion

In this study, we examined the longitudinal dynamics of anti-RBD IgG titers after the second and third (booster shot) doses of the BNT162b2 vaccination in HCWs without previous or de novo infection of SARS-CoV-2 in Japan. Anti-RBD IgG antibody titers peaked at around 1 M after the two-dose regimen vaccination of BNT162b2, followed by a substantial reduction in the IgG levels over time, and anti-RBD IgG titers dropped to nearly one-tenth by 6 M after the second dose of vaccination. We found that two-dose BNT162b2 vaccination-induced anti-RBD IgG responses at the peak (2D-1M) were lower in males than in females. Additionally, antibody titers were inversely correlated with age. Additionally, those who were obese with a BMI of 30 or more had higher antibody titers than those who were not. These results are compatible with the previous reports on the effects of gender and age on humoral responses to the BNT162b2 vaccination [17,18,19,20]. Our cohort individuals received the third dose of the BNT162b2 vaccination at around 8M after the second dose of the BNT162b2 vaccination, in accordance with the Japanese Ministry of Health, Labor and Welfare recommendation to healthcare workers. We found that anti-RBD IgG antibody responses drastically improve after a third-dose vaccination with no major adverse events.

One of the notable findings in this study is the disappearance of effects of age and the male gender on anti-RBD IgG antibody responses after the third dose. We showed that the group of individuals whose anti-RBD IgG responses at 2D-1M were mild, namely, less than mean minus 1 SD, drastically enhanced anti-RBD IgG after a third dose of the BNT162b2 vaccination, as shown in Figure 5B. Accordingly, the negative effects of age and the male gender on anti-RBD IgG antibody titers were no longer observed at 3D-1M (Figure 6C and Figure 7A). Multivariate analysis on effect of sex, age and BMI on anti-RBD IgG responses at 1 M after the second dose revealed that age (−0.230, *p* < 0.001) contributed the most in 2D-1M, followed by the male gender (−0.157, *p* = 0.004) and BMI (0.106, *p* = 0.054). Similar multivariate analysis at 3D-1M revealed that BMI made a stronger (0.134, *p* = 0.017), the male gender made a weaker (0.096, *p* = 0.083) and age made little or no (−0.044, *p* = 0.412) contributions, respectively. Thus, these results strongly suggest that the third dose (booster shot) of the BNT162b2 vaccination strikingly augments anti-RBD IgG responses of those mild responders, namely, older persons and males, which are in line with the recent report by Eliakim-Raz et al., documenting that a third BNT162b2 dose in adults aged 60 years and older was associated with significantly increased anti-SARS-CoV-2 Spike IgG [21]. Collectively, our results confirm key groups that may benefit from additional vaccine doses when available [22].

Our cohort HCWs, given their younger age and relatively good health, achieved higher initial antibody levels after the second dose of BNT162b2 vaccine; however, they experienced significant declines in humoral immunity over time. Anti-RBD IgG titers were reduced in all populations and the half-life of antibody titers at 2D-3M from the peak of 2D-1M was approximately 37 days (Figure 4C). It is worthwhile to mention that the decrease from the start of 2D-3M onward was much slower and the half-life of antibody titers at 2D-6M from 2D-3M was approximately 73 days, indicating the persistence of the antibody for a long period, over the 6 months. This decrease in antibody-decay speed over time was consistent across our cohort, regardless of gender or age. Levin et al. also reported the decrease in neutralizing antibody titers was initially brisk, in the period of up to 70 to 80 days, but slowed thereafter [9]. These results agree with the recent report by Choi J.H. et al., showing the maintenance of SARS-CoV-2 antibody responses in healthcare workers in South Korea 6 months after receiving a second dose of the BNT162b2 mRNA vaccine [23]. This long-term persistence of antibodies, with the assumed contribution of memory B cells and long-lived plasma cells, may explain, in part, why the mRNA vaccine remains highly effective in preventing severe disease, even after the decline of antibody titers over time and increase in mild breakthrough infections. McIntyre P.B. et al. recently reported that COVID-19 vaccine strategies must remain focused on severe disease, and that global equity in achieving high adult coverage of at least one dose is key to minimizing severe COVID-19 [24]. The implementation of booster doses beyond those at highest risk by countries with abundant vaccine supplies could compromise vaccine availability at the global level [24]. Thus, we suggest that further studies would be needed to determine whether a further additional dose of the vaccine is required for all adults, including healthy young adults.

Since several countries, including Japan, have developed policies in favor of a third dose (booster shot) of vaccination, a large proportion of our cohort of HCWs also received a third dose of the BNT162b2 mRNA vaccine at 8 M after the second dose of the vaccine. Thus, we cannot pursue the persistence of antibodies any longer than 6 months after the second dose of vaccination. However, this current study is an ongoing prospective observational study of HCWs who received three doses of BNT162b2 and the future follow-up study of anti-RBD IgG antibody titers at 3D-3M, 3D-6M and 3D-12M will provide very important clues to judge the necessity and the timing of further doses of vaccinations, if needed.

One of the limitations of this study is that the neutralizing activity was not measured in this study. Neutralizing assays are complex and currently impossible to perform in our facility. However, it has been reported that neutralizing responses were directly correlated to IgG anti-RBD titers [25]. A strong correlation between IgG and neutralizing antibody titers was reported to be maintained throughout the 6 months after the receipt of the second dose of vaccine [9]. Thus, we have not examined neutralizing antibody titers in this study. Caution should be exercised when using the antibody titer measured by ELISA to estimate the anti-infection effect against mutant strains. Furthermore, it should be noted that the efficacy of vaccines in preventing the onset and severity of the disease cannot be solely explained by serum antibody titers, including neutralizing activity [26]. T-cell response persists for at least six to eight months and that B-cell-mediated immunity can be sustained at least 12 months after the initial infection [27]. It may be possible to speculate that many individuals may have sufficiently preserved immunity after two-dose vaccinations to prevent severe disease, although further future study is required to determine the longevity of protection [22]. Finally, since our cohort was mainly composed of healthy individuals under 65 years of age, the results of this study cannot be applied to a population that includes older people or those with underlying diseases that may cause immunodeficiency.

## 5. Conclusions

Overall, our study shows that a third dose of BNT162b2 strikingly augments the anti-RBD IgG responses of all the participants and the enhancements of antibody responses were remarkable, especially in the mild responders after the second dose of the BNT162b2 vaccine. The negative effect of age and the male gender on anti-RBD IgG responses to BNT162b2 observed after the second dose of the BNT162b2 vaccine were lost 1 M after the third dose of the BNT162b2 vaccination. On the other hand, peak antibody titers at 1 M after the second and third doses of the vaccine were significantly higher in obese subjects with BMI ≥ 30. Based on the rapid spread of the new variant and reports of vaccine-breakthrough infections, our current results showing a significant reduction in anti-RBD IgG antibody titers 6 to 7 M after the second dose of the vaccine might warrant boosting the immunity of the aged individuals, especially older males. This study confirms the key groups that may benefit from additional vaccine doses when available, and suggests that the third dose (booster shot) of the BNT162b2 vaccine strikingly augments anti-RBD IgG responses of these mild responders and increases vaccine effectiveness to prevent SARS-CoV-2 infection.

## Figures and Tables

**Figure 1 vaccines-10-00830-f001:**
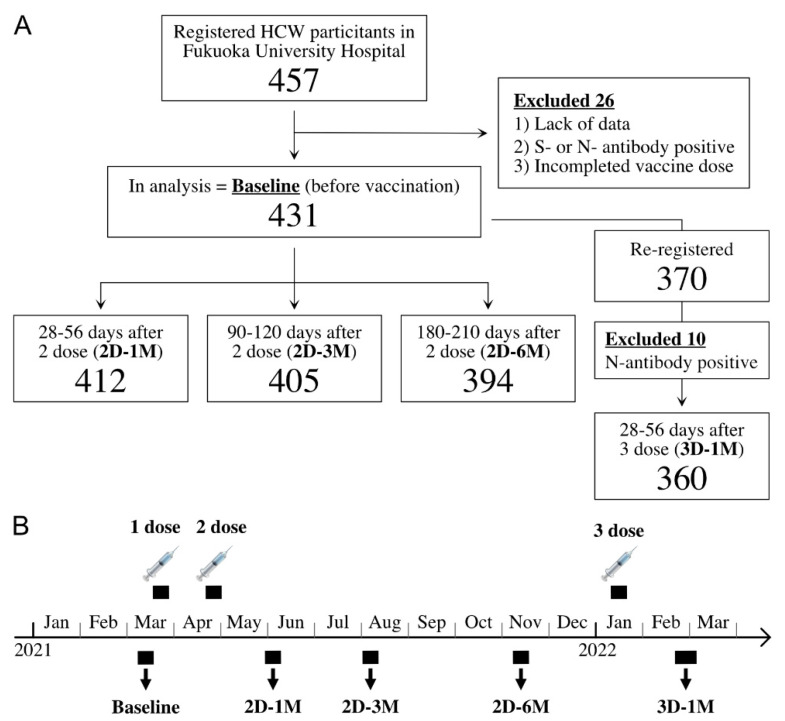
Cohort diagram and schedules of BNT162b2 vaccinations and blood samplings. (**A**) Flowchart of the study cohort, showing the number of participants. This is a prospective observational study of HCWs who received two doses of the BNT162b2 vaccine followed by a third dose of BNT 162b2 with 8 M intervals. Participants were followed three times after the second dose: 2D-1M (28–56 days), 2D-3M (90–120 days) and 2D-6M (180–210 days). Blood samplings were also collected at 1 M after the third dose of BNT162b2 vaccination from the participants (3D-1M). (**B**) BNT162b2 vaccinations and blood collection schedule.

**Figure 2 vaccines-10-00830-f002:**
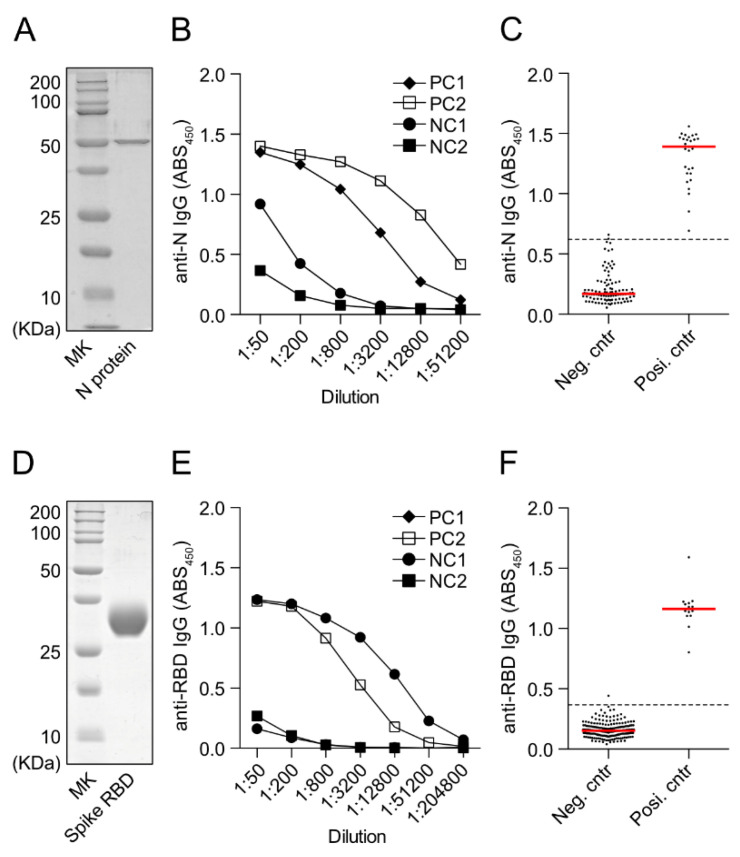
Measurement of anti-SARS-CoV-2 N-protein IgG antibody and anti-SARS-CoV-2 S-protein RBD IgG antibody by enzyme-linked immunosorbent assay (ELISA). (**A**) SDS-PAGE of purified recombinant SARS-CoV-2 N protein (0.4 μg). (**B**) Initial set-up ELISA to detect anti-N-protein IgG using serially diluted COVID-19 patients serum samples (PC1, PC2) and negative controls (NC1: pooled serum, Cosmo bio KOJ-12181201, NC2: clinical sample obtained before the COVID-19 epidemic). (**C**) Serum reactivity of control (*n* = 100) and COVID-19 patients (*n* = 29) to SARS-CoV-2 N protein. ABS450 values of 400-fold diluted serum are shown. Control serum was collected before SARS-CoV-2 spread in Japan and serum samples of COVID-19 patients were collected later than 14 days after the disease onset. (**D**) SDS-PAGE of recombinant SARS-CoV-2 RBD protein (5 μg). (**E**) Initial set-up ELISA to detect anti-RBD IgG using serially diluted COVID-19-patient serum samples (PC1, PC2) and two negative controls (NC1, NC2). (**F**) Serum reactivity of control (*n* = 199) and COVID-19 patients (*n* = 14) to SARS-CoV-2 RBD protein. ABS450 values of 200-fold diluted serum are shown. Red lines indicate median.

**Figure 3 vaccines-10-00830-f003:**
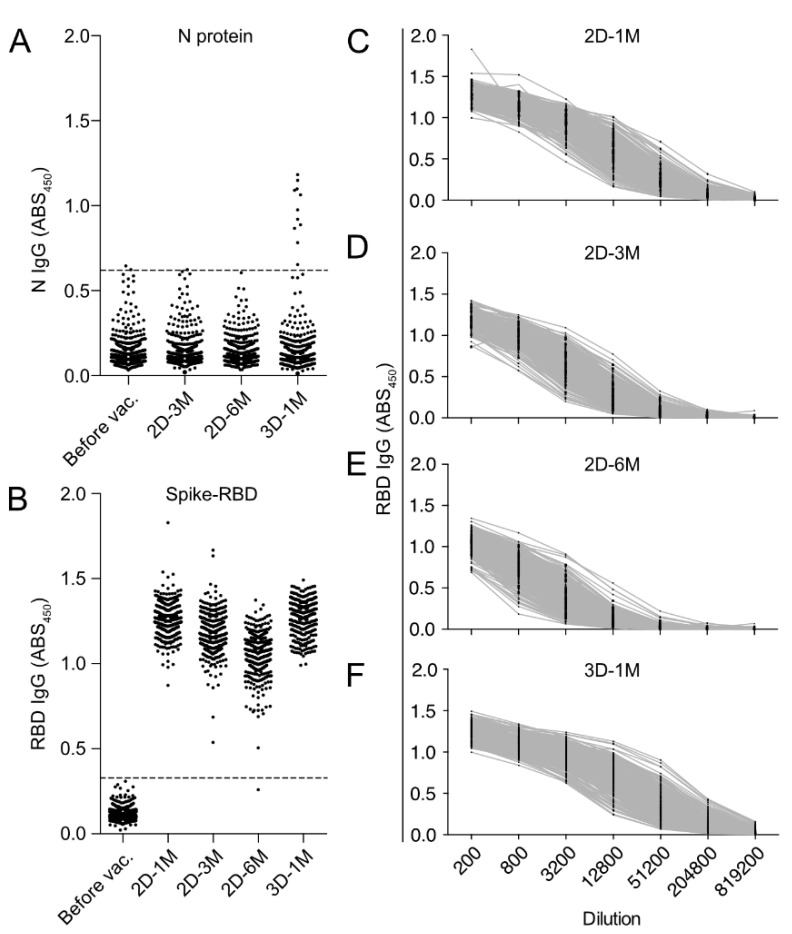
Humoral responses to SARS-CoV-2 N protein and RBD after second and third doses of BNT162b2 mRNA vaccine in HCWs in Japan. (**A**) Anti-SARS-CoV-2 N-protein IgG of serum samples obtained before the vaccination, at 2D-1M (28–56 days), 2D-3M (90–120 days) and 2D-6M (180–210 days) after the two doses of BNT162b2 vaccine, and 3D-1M (28–56 days) after the three doses of BNT162b2 vaccine. ABS450 values of ×400 diluted serum are shown. The dotted line indicates the cutoff for positivity. (**B**) Anti-SARS-CoV-2 RBD IgG from serum samples obtained before the vaccination, 1 M, 3 M and 6 M after the second vaccination, and 1 M after the third vaccination. ABS450 values of ×200 diluted serum are shown. (**C**–**F**) Human-IgG reactivity to SARS-CoV-2 RBD of serially diluted serum obtained at 1 M (*n* = 412), 3 M (*n* = 405) and 6 M (*n* = 394) after the second dose of vaccination and 1 M (*n* = 360) after the third dose of vaccination. Our cohort received the third mass vaccination of BNT162b2 after 8 months after the second dose of BNT162b2 vaccine.

**Figure 4 vaccines-10-00830-f004:**
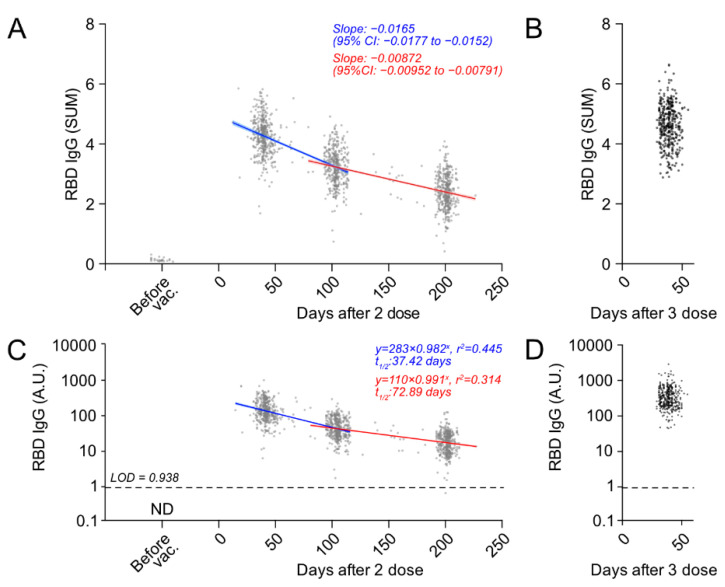
Longitudinal kinetics of anti-SARS-CoV-2 RBD IgG antibodies after the second and third doses of the BNT162b2 vaccine. (**A**) Linear regression analysis of a total of 1252 samples of post-vaccination from 431 subjects. Antibody titers (SUM) of the pre-vaccination phase from randomly selected 24 participants are also shown. The sum of ABS_450_ values of six serial 4-fold dilutions (SUM) from each serum sample are analyzed. Linear regression models are fitted to the data from the peak humoral response (1 M) to 3 M (blue) and the data from 3 M to 6 M (red), respectively. (**B**) Anti-RBD IgG antibody titers (SUM) at 1 M after a third dose of BNT162b2 (3D-1M) are shown. (**C**) Antibody titers (A.U.) of each sample are determined by standard human IgG monoclonal antibody against SARS-CoV-2 S1 protein. The half-life of antibody titer is calculated from the regression coefficients (37.42 days (95% CI: 34.80–40.46) and 72.89 days (95% CI: 66.23–81.02), respectively). Antibody titers of pre-vaccination phase from randomly selected 24 participants were under the limit of detection. LOD: limit of detection. (**D**) Anti-RBD IgG antibody titers (A.U.) at 1 M after a third dose of BNT162b2 (3D-1M) were shown.

**Figure 5 vaccines-10-00830-f005:**
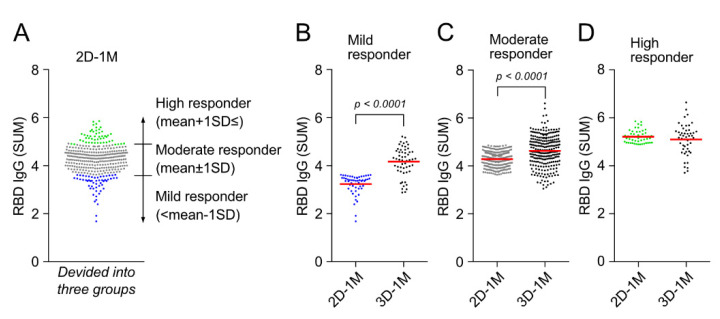
Third dose of BNT162b2 vaccination augments anti-RBD IgG responses of mild responders at the peak of the second dose of BNT162b2 vaccination. (**A**) Cohort participants are divided into three groups based on anti-RBD IgG titers (SUM) at 2D-1M, namely, mild responder: <mean −1 SD (expressed in blue, moderate responder: mean ± 1 SD (expressed in black) and high responder: mean + 1 SD≤ (expressed in green). (**B**–**D**) Anti-RBD IgG titers at 1 M after the third dose of BNT162b2 vaccination (3D-1M) of mild (**B**), moderate (**C**) and high responders (**D**) determined at 2D-1M were compared with the titer of 2D-1M in these three groups.

**Figure 6 vaccines-10-00830-f006:**
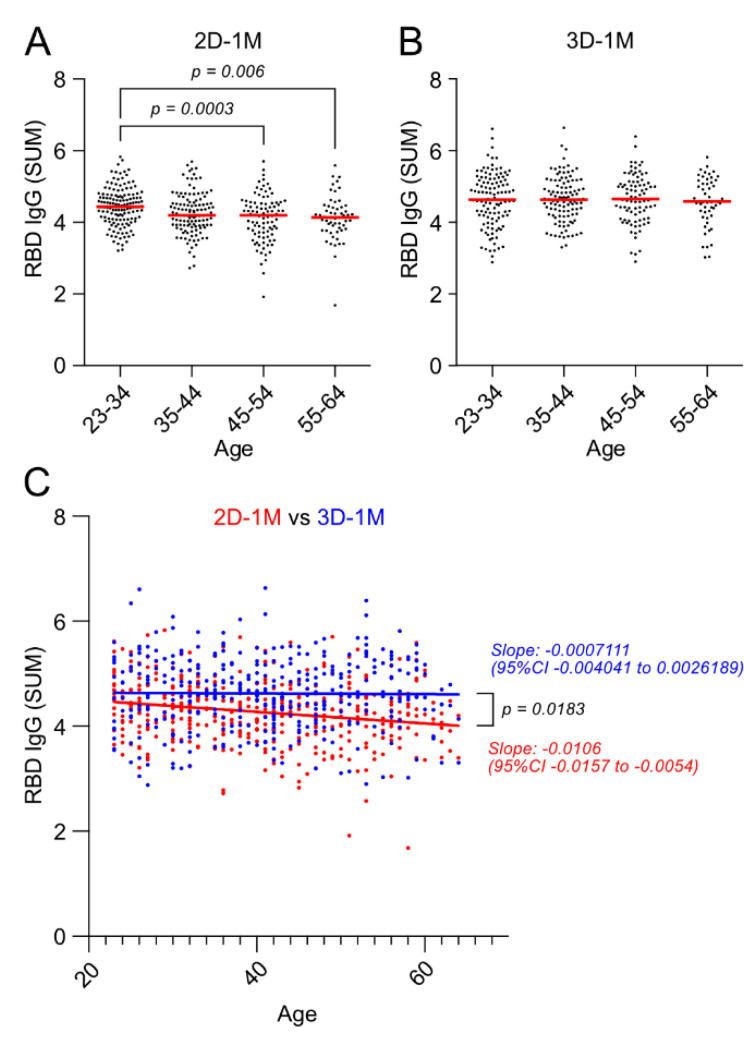
Negative effect of age on anti-RBD IgG responses after the second dose of BNT162b2 was lost after the third dose of BNT162b2 vaccination. Anti-RBD IgG are measured using serial dilutions of serum obtained from participants in an indirect ELISA assay and the sum of ABS450 values of six serial 4-fold dilutions (SUM) is calculated. (**A**,**B**) Effect of age on anti-RBD IgG responses 1 M after the second (2D-1M) and third (3D-1M) doses of BNT162b2 are analyzed by age groups (in 10-year bins) of the HCW cohort. (**C**) Regression analysis of RBD IgG with age of individuals is shown. RBD IgG titers of individuals obtained at 2D-1M are shown in red and individual RBD-IgG titers at 3D-1M are shown in blue. Linear regression lines obtained from 2D-1M and 3D-1M are statistically significantly different (*p* = 0.0183).

**Figure 7 vaccines-10-00830-f007:**
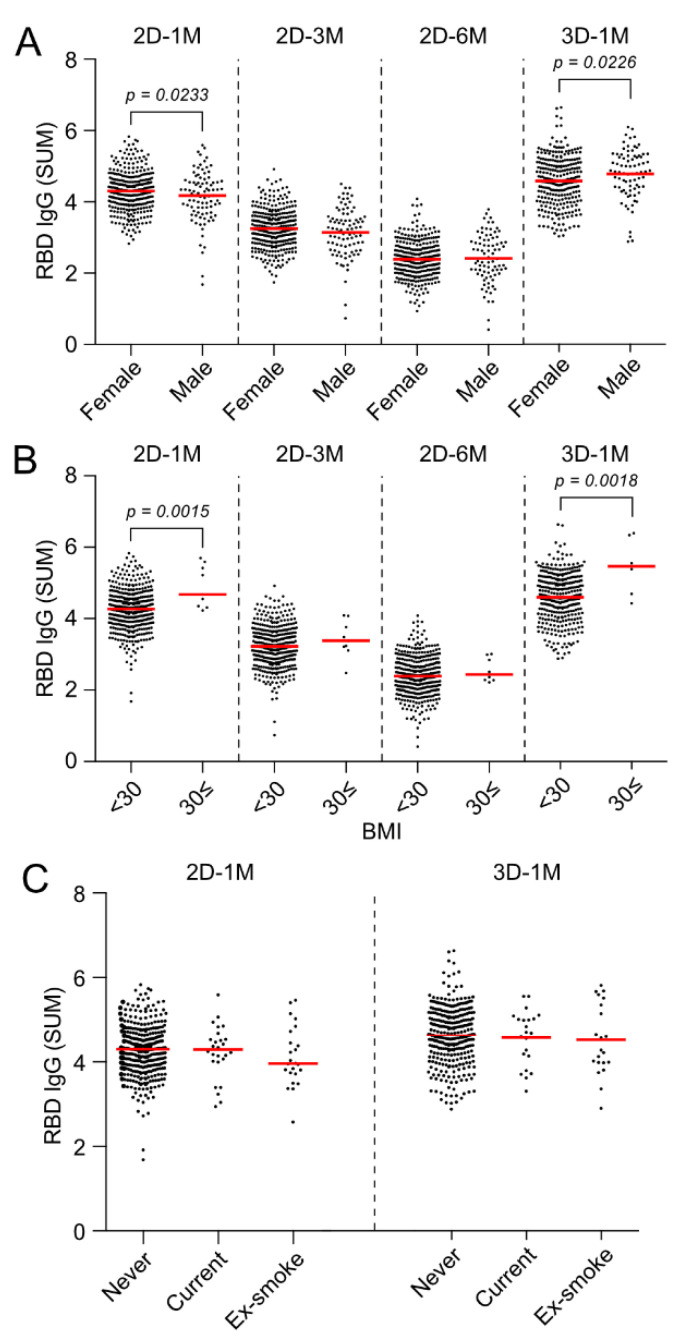
Effect of gender and BMI on anti-RBD IgG responses at 3 M and 6 M after the second dose and 1 M after the third dose of BNT162b2 vaccinations. (**A**,**B**) Effect of gender difference (**A**) and BMI (**B**) on anti-SARS-CoV-2 RBD IgG titers (SUM) at 3 M and 6 M after the second dose and 1 M after the third dose of BNT162b2. Horizontal lines represent the median. (**C**) Effect of smoking status on anti-SARS-CoV-2 RBD IgG titers (SUM) at 1 M after the second dose (2D-1M) and 1 M after the third dose (3D-1M) of BNT162b2. The horizontal lines in (**A**–**C**) are the median.

**Table 1 vaccines-10-00830-t001:** The demographic characteristics of the participants.

Sex			BMI		
Female	337	(78.2%)	<18.5	62	(14.4%)
Male	94	(21.8%)	18.5≤, <25	313	(72.6%)
Age			25≤, <30	43	(10.0%)
			30≤	9	(2.1%)
23–34	150	(34.8%)	NA	4	(0.9%)
35–44	126	(29.2%)	Smoking status		
45–54	95	(22.0%)	Never smoker	378	(87.7%)
55–64	58	(13.5%)	Current smoker	28	(6.5%)
NA	2	(0.5%)	Ex-smoker	25	(5.8%)

**Table 2 vaccines-10-00830-t002:** Characteristics of three groups based on anti-RBD IgG responses at 2D-1M (mild, moderate high responders).

	Mild Responder		Moderate Responder		High Responder	
*n*	53		249		50	
Sex						
Female	33	(62.3%)	203	(81.5%)	39	(78.0%)
Male	20	(37.7%)	46	(18.5%)	11	(22.0%)
Age						
median	45.5		41		40	
IQR	36.0–52.0		32.0–41.0		28.3–40.0	
BMI						
median	21.5		20.7		21.8	
IQR	20.0–22.7		19.1–23.0		19.9–24.1	
Smoking						
never	43	(81.1%)	224	(90.0%)	45	(90.0%)
current	5	(9.4%)	13	(5.2%)	2	(4.0%)
ex-	5	(9.4%)	12	(4.8%)	3	(6.0%)

**Table 3 vaccines-10-00830-t003:** Multiple regression analysis of anti-RBD IgG titers (SUM) in 2D-1M and 3D-1M (standard partial regression coefficients are in parentheses).

	2D-1M	3D-1M	
	All	All	Selected ^1^
(Intercept)	4.380 ***	4.037 ***	3.958 ***
	*p* < 0.001	*p* < 0.001	*p* < 0.001
Age	−0.013 (−0.230) ***	−0.003 (−0.044)	
	*p* < 0.001	*p* = 0.412	
Sex	−0.230 (−0.157) **	0.157 (0.096) ^+^	0.158 (0.096) ^+^
	*p* = 0.004	*p* = 0.083	*p* = 0.080
BMI	0.022 (0.106) ^+^	0.031 (0.134) *	0.029 (0.127) *
	*p* = 0.054	*p* = 0.017	*p* = 0.022
Num. Obs.	346	346	346
R^2^	0.074	0.034	0.032
R^2^ Adj.	0.066	0.025	0.026
AIC	621.1	716.0	714.7
RMSE	0.59	0.68	0.67
F	9.166	3.967	5.619
*p*	<0.001	0.008	0.004

^+^ *p* < 0.1, * *p* < 0.05, ** *p* < 0.01, *** *p* < 0.001; ^1^ Based on AIC, R^2^ (R-square), R^2^ Adj. (adjusted R-square), AIC (Akaike’s Information Criterion), RMSE (Root-Mean-Square Error).

## Data Availability

The data presented in this study are available on request from the corresponding author. The data are not publicly available according to the ethical committee decision the conduct of this study.

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
