# Peer review of "Longitudinal Dynamics of SARS-CoV-2 IgG Antibody Responses after the Two-Dose Regimen of BNT162b2 Vaccination and the Effect of a Third Dose on Healthcare Workers in Japan"

_vaccines, 2022, doi:10.3390/vaccines10060830_

Round 1
Reviewer 1 Report
Major comments.
This paper describes the study on the longitudinal dynamics of SARS-CoV-2 IgG antibody responses after the two-dose regimen of BNT162b2 (Pfizer-BioNTech) vaccination in health care workers in Japan. Further, the authors describe the effects of a third dose of the BNT162b2 vaccine on the antibodies. They found that older persons and males showed lower anti-RBD IgG responses than younger adults and females, respectively, after the two-dose regimen of the vaccination. They found that the decay in anti-RBD IgG titers started from 1 M after the second dose of BNT162b2 and the anti-RBD IgG titers dropped to nearly one-tenth at 6 M after the second vaccination. However, the participants who received a third dose of BNT162b2 vaccine showed significant increases in anti-RBD antibody (17-fold of the titers of those at 6M after the second dose and 2.2-fold of the peak antibody titers at 1 M after the second dose). Most importantly, the negative effect of age and male gender on the anti-RBD IgG response after the second dose of the BNT162b2 vaccine were lost after the third dose of the vaccination. These remarkable effects of the BNT162b2 vaccination on the anti-RBD IgG titers and the detailed analyses conducted by the authors on the longitudinal dynamics of the responses of anti-RBD IgG are very significant and these results would provide useful contribution. Accordingly, the present study is worthy of publication in “vaccine”, if the authors can clarify following points.
- According to the cohort diagram (Fig. 1A), 26 individuals are excluded from the registered 457 participants. As the reasons for the exclusion, the authors pointed three cases; (1) lack of data, (2) S- or N-antibody positive, (3) in-completed vaccine dose. Are there any apparent SARS-CoV-2-infections being occurred among the registered 457 participants during the longitudinal study? Or such infected individuals are included in case (2)? If so, the authors should show the number.
- Being related to the point 1, as the stage of the longitudinal study being progressed, the number of participants decreased from 431 to 412 (2D-1M), 405 (2D-3M), and 394 (2D-6M). It might be better to show the reasons of the decrease in number, as for the first exclusion.
- As for the reason of the exclusion of 10 individuals for the study on the third-dose, only “N-antibody positive” is reasoned. In this case, any S-antibody positive exist?
- In the manuscript, the authors concluded that “The negative effect of age and male gender on anti-RBD IgG responses to BNT162b3 observed after the second dose of BNT162b2 vaccine were lost 1M after the third dose of BNT162b2 vaccination” (page 14, lines 483-486). As a reviewer, I agreed with this conclusion. However, in the abstract, only the negative effect of age was mentioned. Why?
Minor comments.
- “RBD” should be spelled out in its first appearance in the main text (page 1, line 39).
- “SUM calculation” is used without definition (page 3, line 104). Does SUM stand for “the absorbance sum across 6 additional fourfold serial dilutions”?
- In Figure 2A an B, amounts of the loaded protein (in µg) in SDS-PAGE should be
- In Figure 1 panel A, “Re-resistered” should be “Re-registered”.
- “sum” should be “SUM” (page 11, line 355).
- “3M-1M” should be “3D-1M” (page 12, line 369).
- “3M-1M” should be “3D-1M” (page 13, line 409).
Reviewer 2 Report
In the present paper the authors longitudinally evaluated the dynamics of IgG antibody responses against SARS-CoV-2 after the two-dose regimen of BNT162b2 vaccination and the Effect of a third dose in health care workers in Japan. They selected 431 COVID-19 naïve health care worker donors and measured anti-RBD IgG at 1, 3 and 6 months after the second dose of BNT162b2 and at one month after a third dose of BNT162b2 vaccination. They found that all donors mounted high anti-RBD IgG responses after the two-dose regimen of BNT162b2 vaccination. Older persons and males presented lower anti-RBD IgG responses compared to younger adults and females, respectively. The decline in anti-RBD IgG titers was observed after one month post second dose of BNT162b2 and titers dropped to nearly 10% at six months post second vaccination. Interestingly after a third dose of BNT162b2 given at eight months post-second dose, anti-RBD antibody titers dramatically increased to reach higher values than the peak antibody titers at 1 M after the second dose of vaccination. They concluded that these results suggest that the third dose of BNT162b2 safely improves humoral immunity against SARS-CoV-2 with no major adverse events.
General comments:
This paper is well written easy to read, and the data are explicitly presented. The written English is very good, and the text is well organized. The authors recognize that the weakness is the lack of evaluation of neutralizing antibodies which are the main components of the humoral response that correlate with the protection.
Specific comments:
Figure 1B there is anti-N IgG in NC-1 (1:50 and 1:200) but lower anti-S? No comment was addressed for this discrepancy! Is this increased response linked to cross reactivity with the N of common-cold coronaviruses?
Figure 1C there are a high proportion of samples in the negative control group that have titers of anti-N IgG at dilution 1:400 that higher than NC-1 and NC-2! Any comment on this?
Fig 3A 3D-1M high anti-N responses in ten participants. Are these resulting from new infections with different variants after two doses vaccination? Were these samples included in the 3B data analysis?
Since there was no balanced sex-ratio; there might be a bias on the data of the effect of gender and BMI on IgG responses at 3, 6 months after the second shot & 1 month after the 3rd !?
Fig 5 legend the last sentence needs to be corrected “mild responder (B), Moderate responder ( C) and high responders (D)”
It is not clear why the third dose of vaccination in high responders failed to increase further anti-RBD titers while it did increase the responses in moderate and mild responders.
Reviewer 3 Report
The manuscript by Sakamoto et al. reports the analysis of longitudinal dynamics of humoral response to the BNT162b2 vaccine in a cohort of naïve COVID-19 health care workers in Japan. Measurements of humoral response were carried out at about 1, 3, and 6 months after the 2nd vaccine dose and, for a slightly smaller pool, 1 month after the booster shot.
On overall, I consider this manuscript well written and sound from a scientific point of view. Of note, the study is carried out on a large pool of individuals, affording good statistical significance to results. Results agree with previous studies, which identifies age, and to a lower extent sex, as a relevant factor to Ab level after the 2ndshot. The authors correctly suggest that, in a desirable context of equal access to vaccine worldwide, the booster shot should be made available with priority to individuals with recognized lower immunity after 2ndshot, including elderly individuals. Yet, and this in my opinion is the main issue of the manuscript, the authors fail to produce a “kinetic” analysis along time stratified by age. Indeed, I think the authors should add to the manuscript:
- An age (and perhaps sex) analysis on the individual propensity to Ab decline, calculating the ratio Ab(3M)/Ab(1M) and Ab(6M)/Ab(3M) for each individual and evaluating its dependence (multivariate regression) on the variables above.
In my opinion, this analysis would greatly add value to the scientific message, given the excellent time sampling of the study, and it would afford information on the age (and possibly sex) propensity to Ab decline over time, which is poorly known. For this reason, I suggest "major revision", although my suggestion is not out of scientific flaws, but to add value to the scientific message.
Other minor issues
- Sentence starting with “On the other hand…” at line 49 must be split into two sentences, with the second starting with “Yet,” and substituting “that antibody” with “the antibody”
- “we next moved…” instead of “we next move” at line 290
- The paragraph between 449 and 462 is a repetition of already presented concepts, and I would remove it to improve manuscript readability
Round 2
Reviewer 3 Report
The authors satisfactorily addressed all my comments.